# Gain-of-Function Dynamin-2 Mutations Linked to Centronuclear Myopathy Impair Ca^2+^-Induced Exocytosis in Human Myoblasts

**DOI:** 10.3390/ijms231810363

**Published:** 2022-09-08

**Authors:** Lucas Bayonés, María José Guerra-Fernández, Fernando Hinostroza, Ximena Báez-Matus, Jacqueline Vásquez-Navarrete, Luciana I. Gallo, Sergio Parra, Agustín D. Martínez, Arlek González-Jamett, Fernando D. Marengo, Ana M. Cárdenas

**Affiliations:** 1Instituto de Fisiología, Biología Molecular y Neurociencias, CONICET, Departamento de Fisiología y Biología Molecular y Celular, Facultad de Ciencias Exactas y Naturales, Universidad de Buenos Aires, Buenos Aires 1428, Argentina; 2Centro Interdisciplinario de Neurociencia de Valparaíso, Facultad de Ciencias, Universidad de Valparaíso, Gran Bretaña 1111, Valparaíso 2360102, Chile; 3Centro de Investigación de Estudios Avanzados del Maule (CIEAM), Vicerrectoría de Investigación y Postgrado, Universidad Católica del Maule, Talca 3460000, Chile; 4Centro de Investigación en Neuropsicología y Neurociencias Cognitivas (CINPSI Neurocog), Facultad de Ciencias de la Salud, Universidad Católica del Maule, Talca 3460000, Chile; 5Instituto de Fisiología Celular—Neurociencias, Universidad Nacional Autónoma de México, Mexico City 04510, Mexico; 6Escuela de Química y Farmacia, Facultad de Farmacia, Universidad de Valparaíso, Valparaíso 2360102, Chile

**Keywords:** dynamin, dynamin-2 mutations, centronuclear myopathy, endocytosis, exocytosis, GLUT4, pHluorin, IRAP

## Abstract

Gain-of-function mutations of dynamin-2, a mechano-GTPase that remodels membrane and actin filaments, cause centronuclear myopathy (CNM), a congenital disease that mainly affects skeletal muscle tissue. Among these mutations, the variants p.A618T and p.S619L lead to a gain of function and cause a severe neonatal phenotype. By using total internal reflection fluorescence microscopy (TIRFM) in immortalized human myoblasts expressing the pH-sensitive fluorescent protein (pHluorin) fused to the insulin-responsive aminopeptidase IRAP as a reporter of the GLUT4 vesicle trafficking, we measured single pHluorin signals to investigate how p.A618T and p.S619L mutations influence exocytosis. We show here that both dynamin-2 mutations significantly reduced the number and durations of pHluorin signals induced by 10 μM ionomycin, indicating that in addition to impairing exocytosis, they also affect the fusion pore dynamics. These mutations also disrupt the formation of actin filaments, a process that reportedly favors exocytosis. This altered exocytosis might importantly disturb the plasmalemma expression of functional proteins such as the glucose transporter GLUT4 in skeletal muscle cells, impacting the physiology of the skeletal muscle tissue and contributing to the CNM disease.

## 1. Introduction

Centronuclear myopathy (CNM) is a group of congenital ailments that mainly affect the skeletal muscle tissue. They are caused by mutations in genes encoding proteins involved in membrane trafficking and remodeling (dynamin-2, amphiphysin-2 and myotubularin), sarcomere stabilization (titin) or Ca^2+^ homeostasis (ryanodine receptor channel type 1) [1]. They are characterized clinically by progressive weakness and skeletal muscle atrophy and histologically by the presence of myofibers with centrally placed nuclei [1]. Around 15% of CNM cases are associated with dynamin-2 mutations [2], which cause heterogeneous clinical manifestations from mild late-onset to severe neonatal-onset forms [3]. Dynamin-2 is a ubiquitously expressed enzyme belonging to the dynamin superfamily, a group of large GTPases that self-assemble to exert their mechanoenzyme activity [4]. They comprise five functional domains: an N-terminal GTPase domain that hydrolyzes GTP, middle and GTPase effector domains that are involved in self-assembly, a pleckstrin homology domain (PH) that binds to membranes, and a C-terminal proline-rich domain (PRD) that interacts with SH3-domain-containing partners [4]. These GTPases are involved in endocytosis, caveolae internalization and membrane budding from trans-Golgi networks, as they catalyze the scission of nascent vesicles from the plasmalemma or Golgi cisterna membranes by assembling into helices that surround the neck of nascent vesicles [5]. Dynamins also regulate the dynamics of the actin cytoskeleton in different types of cells [6,7,8,9,10]. Furthermore, they are involved in exocytosis, a cellular process that allows the fusion of membranous organelles, mainly transport and secretory vesicles, with the plasmalemma [8,11,12,13,14,15]. It has been proposed that during exocytosis, dynamins facilitate vesicle fusion with the plasmalemma [11], as well as promote the closure of the fusion pore (a temporary structure formed during exocytosis), giving rise to a mode of exocytosis known as kiss-and-run [14,16,17]. In the skeletal muscle, exocytosis is an essential process that allows the insertion of functional proteins, such as the glucose transporter GLUT4, into the plasmalemma [18] and the secretion of myokines, cytokines or other peptides that are produced and released by skeletal muscle cells in response to contraction [19].

Dynamin-2 is the only reported dynamin isoform in skeletal muscle tissues, where it has been shown to be involved in endocytosis, cytoskeletal actin dynamics, membrane trafficking and remodeling, and myofiber development [10,20,21,22,23,24]. Then, dynamin-2 malfunction might disrupt several cellular processes, including exocytosis. In this regard, it has been reported that impaired GLUT-4 insertion into the plasmalemma of human myoblasts and mice myofibers expressing the dynamin-2 p.R465W mutation causes mild late-onset CNM and aberrant perinuclear accumulation of this glucose transporter in biopsies from CNM patients carrying the p.R465W mutation [21]. Considering that exocytosis is a pivotal mechanism that has not been investigated enough in skeletal muscle cells, we studied in the present work how the gain-of-function dynamin-2 mutations p.A618T and p.S619L [25] influence this process. Both p.A618T and p.S619L mutations cause a neonatal myopathic phenotype with severe clinical manifestations [3]. They are located in the C-terminal α-helix motif of the PH domain, a region that seems to connect the association to membranes with the GTPase activity [25]. Both mutations disturb such coupling, with the p.A618T mutant substantially enhancing the GTPase activity in the presence of lipids and the p.S619L mutant exhibiting an elevated basal GTPase activity that is not increased by lipids [25]. With this question in mind, we directly measured single exocytotic events using total internal reflection fluorescence microscopy (TIRFM) in immortalized human myoblasts expressing IRAP (an insulin-responsive aminopeptidase that colocalizes with GLUT4 in vesicles) tagged with the pH-sensitive green fluorescent protein (pHluorin) [26]. With this experimental approach, we demonstrated that the gain-of-function of dynamin-2 impairs exocytosis and consequently might impact the expression of functional proteins at the plasmalemma of skeletal muscle cells.

## 2. Results

### 2.1. pHluorin Signals in Human C25 Myoblasts

Regulated exocytosis is a Ca^2+^-dependent process mediated by the SNARE (from the acronym Soluble N-ethylmaleimide-sensitive factor Attachment protein Receptor) proteins. They assemble into a helix complex that brings the vesicle membrane and plasmalemma closer together, promoting their fusion [27]. Stimuli, such as muscle contraction and insulin, promote Ca^2+^-dependent exocytosis in skeletal muscle cells [28,29]. In the present work, we evaluated whether ionomycin, a Ca^2+^ ionophore that induces transient increments of cytosolic Ca^2+^ concentrations [30] and increases plasmalemma GLUT4 expression in myoblasts [21,31], promotes exocytosis in the immortalized human myoblasts C25. Ca^2+^ signals induced by 1, 5, and 10 μΜ ionomycin are shown in the Appendix A. Therefore, we measured the exocytosis of vesicles containing IRAP-pHluorin in response to the application of these three different concentrations of ionomycin. pHluorin is a pH-sensitive fluorescent protein that is quenched at the acidic pH of vesicles, but during exocytosis, its fluorescence drastically increases when exposed to the extracellular medium (pH = 7.4). Then, every single exocytotic event is observed as a bright fluorescent spot that spreads laterally as the vesicle proteins diffuse across the plasmalemma or that dims without lateral spreading as the vesicle proteins are retrieved at fusion sites [32,33]. The frequency of both types of pHluorin signals (with lateral and non-lateral diffusion) is shown in the Appendix A. In non-stimulation and 10 µM ionomycin stimulation conditions prevailed the events with non-lateral diffusion. Figure 1A shows a TIRFM image and a selected fluorescence event in a non-stimulated C25 cell. The number of these types of events did not significantly change after the application of 1 μM ionomycin, but concentrations of 5 and 10 μM of this ionophore strongly increased the number of events with respect to the non-stimulated condition (Figure 1B). Ionomycin at 10 µM increased both lateral and non-lateral diffusion events as compared with non-stimulation conditions (Appendix A). Appendix A shows fluorescence events in a C25 cell stimulated with 10 μM ionomycin, and Figure 1C shows an event in a kymograph with the respective trace of this event in time.

We also assayed the effects of insulin on the amount of these fluorescence events and found that insulin at 100 nM increased by 3.5-fold the event frequency with respect to non-stimulated cells (Figure 1D).

To determine whether the pHluorin signals correspond to exocytosis or to the arrival of vesicles with neutral pH to the plasma membrane, cells were stimulated with 10 μM ionomycin in an acidic solution (pH = 5.5), which reportedly quenches the pHluorin molecules exposed to the extracellular medium [34]. Only a few fluorescence events (0.002 ± 0.001 min^−1^ μm^−1^; 4 cells) were observed in the presence under 10 μM ionomycin stimulation in this acidic solution (see Appendix A), being less than 1% of the events observed with 10 μM ionomycin at pH 7.4 extracellular solution (*p* < 0.005), suggesting that the pHluorin signals observed here corresponded to exocytosis. To be sure that the acidic extracellular solution did not change the intraluminal pH of intracellular vesicles, cells expressing IRAP-pHluorin were visualized by confocal microscopy. The presence of fluorescent vesicles in a neutral extracellular solution (pH = 7.4) indicates that some vesicles or organelles have non-acid intraluminal pH (see Appendix A). After changing the extracellular pH from 7.4 to 5.5, such vesicles continued being fluorescent (Appendix A). The analysis of 5 cells yielded a Pearson’s correlation coefficient of 0.85 ± 0.03.

### 2.2. Kinetics of the IRAP-pHluorin Signals in C25 Myoblasts Depends on the Stimulus

At non-stimulated and 1 μM ionomycin-stimulated conditions, most of the events had a duration shorter than the recording time (77 ± 17% and 67 ± 23% of events from 10 and 12 cells, respectively), whereas in 5 and 10 μM ionomycin-stimulated conditions, 85 ± 25% and 90 ± 8.3% of events (10 cells each one) ended after the recording finished. Under insulin stimulation, 76 ± 4% of events from 10 cells had a duration shorter than the recording time. Then, these different types of stimuli seem to also define the time of residence of IRAP-pHluorin vesicles at the plasmalemma. Therefore, we analyzed the duration, dwell-time length and decay time of the events that occurred entirely during the recording period, which we called “transient events”. Figure 2A shows a typical trace of a transient event, including a representation of the three analyzed parameters. Dwell-time corresponds to the time between the end of the initial fast fluorescence increase and the start of fluorescence decay ((2) in Figure 2B). It has been proposed that it reports the residency of vesicle proteins at the plasmalemma before its internalization by endocytosis [35]. It has also been pointed out that it may reflect the duration of the fusion pore before its closing [36]. Depending on the mode of exocytosis, the decay time may reflect the vesicle retrieval and its subsequent reacidification after kiss-and-run or the lateral diffusion of the vesicle protein after full fusion [36,37].

Event durations in the 5 and 10 μM ionomycin-stimulated conditions were 2.5- and 2.9-fold longer than that of spontaneous events (Figure 2B(i)). Dwell-time lengths in cells stimulated with 5 and 10 μM ionomycin also were longer (3.5-fold) than those in non-stimulated cells (Figure 2B(ii)). Finally, as shown in Figure 2B(iii), decay times were not significantly different between the experimental conditions. Mean values of duration, dwell time, and decay time under insulin stimulation were 25.5 ± 2.1 s, 5.0 ± 0.8 s and 27 ± 4 s (*n* = 10), not significantly different from those observed on non-stimulated cells.

We further performed experiments in the presence of 100 mM HEPES in the extracellular solution. If HEPES is able to enter into the vesicular lumen, it would slow the vesicle acidification and, therefore, the decay time [36]. As shown in Appendix A, 100 mM HEPES significantly increased averaged decay values of the pHluorin signals induced with 1 μM ionomycin. Additionally, τ values obtained from the single exponential fits of the survival curves were 4-fold slower in the presence of 100 mM HEPES (Appendix A). These results show that in our experiments, the decay time depends, at least in part, on the vesicle reacidification. More importantly, this further confirms that we are measuring exocytotic events since HEPES, to exert its effects, had to diffuse through an opened fusion pore.

### 2.3. Dynamin-2 p.A618T and p.S619L Mutations Reduce Exocytosis in C25 Myoblasts

As dynamin-2 seems to play an important role in vesicle trafficking, exocytosis and endocytosis [38], we subsequently analyzed how the gain-of-function dynamin-2 mutations p.A618T and p.S619L influence the characteristics of the single fluorescence events. We previously described that the dynamin-2 mutant p.R465W, which also seems to display a gain-of-function activity [20,39], impairs GLUT-4 insertion into myoblast plasmalemma [21]. Therefore, we analyzed whether the dynamin-2 mutants p.A618T or p.S619L also disturb the exocytosis of IRAP-pHluorin induced with 10 μM ionomycin.

The efficiency of the transfection with WT dynamin-2 or the mutants A618T or S619L was low, with only 2.2 ± 0.7, 3.5 ± 0.7 and 2.3 ± 0.7% transfected cells by coverslip (Appendix A). As visualized by confocal microscopy using a 60× objective, all these dynamin-2 variants displayed a homogeneous cellular distribution, and only 8.7%, 12.5% and 9.1% of 23, 24 and 22 cells transfected with dynamin-2 WT, A618T or S619L, respectively, showed cytosolic aggregations.

We first compared the Ca^2+^ signals induced by 10 μM ionomycin in C25 myoblasts expressing the mCherry-tagged plasmid containing WT dynamin-2, the mutation p.A618T or the mutation p.S619L. As shown in Appendix A, the amplitudes of the ionomycin-induced Ca^2+^ signals were not significantly different between these three conditions. However, all of them were significantly lower than those induced in cells transfected with the empty mCherry vector (Appendix A). The Ca^2+^ signals induced in cells overexpressing WT dynamin-2 also had a significantly smaller amplitude than those observed in non-transfected cells (Appendix A).

Next, we analyzed the pHluorin signals in the C25 myoblasts transfected with IRAP-pHluorin and the different mCherry-tagged plasmids. Figure 3A shows confocal images of C25 myoblasts co-expressing IRAP-pHluorin with every dynamin-2 variant. As expected, the number of pHluorin signals per minute significantly increased in response to the application of 10 μM ionomycin by 28-, 17- and 15-fold over the non-stimulated condition in cells expressing WT, p.A618T or p.S619L dynamin-2, respectively. More importantly, while the frequency of spontaneous events was similar between the three dynamin-2 variants, the number of pHluorin signals induced with 10 μM ionomycin in C25 myoblasts expressing the p.A618T and p.S619L mutations was significantly lower than that of the WT condition (Figure 3B).

As in non-transfected cells, non-lateral diffusion events prevailed over lateral diffusion events in the three transfection conditions (Appendix A). Importantly, both types of events were decreased in cells expressing the p.A618T and p.S619L mutations as compared with the WT condition (Appendix A).

Next, we analyzed the durations, dwell-time lengths and decay times of the transient events induced with 10 μM ionomycin in cells expressing p.A618T, p.S619L or WT dynamin-2 (Figure 4). As compared with WT dynamin-2, the p.A618T and p.S619L mutants reduced the length of the events by 60% and 68%, respectively (Figure 4A(i)).

A comparison of the distribution frequency of the individual duration values showed that, different from the WT condition, events with durations over 75 s were almost absent in cells expressing the p.A618T or p.S619L mutations (Figure 4A(ii)). Furthermore, significant differences (*p* < 0.001) between WT and both mutants were obtained for the normalized cumulative distributions of durations (continuous lines Figure 4A(ii)). Moreover, when the inverse of the normalized cumulative distributions (also called survival curves) was fitted with a single exponential decay function (R_WT_ = 0.9986; R_A618T_ = 0.9996; R_S619L_ = 0.9992), the resultant time constants (τ) for p.A618T and p.S619L (4.2 ± 0.1 s and 4.3 ± 0.1 s, respectively) were significantly lower than the τ (4.8 ± 0.2 s) obtained for the WT (Figure 4A(iii)).

Dwell-time lengths were also significantly shorter in the p.A618T and p.S619L mutations as compared with the WT condition (Figure 4B(i)). Again, frequency histograms show that dwell times longer than 40 s were almost absent in the p.A618T and p.S619L conditions (Figure 4B(ii)), also showing significant differences in cumulative histograms when compared with the WT condition. When fitted with a single exponential decay function (R_WT_ = 0.9985; R_A618T_ = 0.9999; R_S619L_ = 0.9996), τ were 4.9 ± 0.2 s, 4.3 ± 0.2 s and 4.5 ± 0.1 s for the WT, p.A618T and p.S619L conditions, respectively, again showing a significant decrease for both mutants in comparison with the WT condition (Figure 4B(iii)).

Finally, the mutants p.A618T and p.S619L accelerated decay times (Figure 4C(i)). One more time, analyses of the cumulative distribution of the individual values showed that decay times longer than 50 s were absent in cells expressing the mutants (Figure 4C(ii)), with significant differences as compared with the WT condition. Furthermore, τ values obtained from the single exponential fits of the survival curves (R_WT_ = 0.9880; R_A618T_ = 0.9995; R_S619L_ = 0.9994) were also significantly smaller for the p.A618T and p.S619L mutations (τ= 5.6 ± 0.2 s and 6.1 ± 0.3 s, respectively), as compared with the cells transfected with WT dynamin-2 (τ= 8.2 ± 0.7, Figure 4C(iii)).

### 2.4. Dynamin-2 p.A618T and p.S619L Mutations Impairs the Formation of Actin Filaments in C25 Myoblasts

Given the fact that the CNM mutation p.R465W impairs the formation of actin filaments in myoblasts [21], and cytoskeletal actin dynamics mediate the action of dynamin-2 on exocytosis in secretory cells [8] and GLUT-4 insertion in myoblasts [21], we analyzed whether the p.A618T and p.S619L mutations affect the formation of de novo actin filaments in C25 myoblasts. In these experiments, myoblasts expressing the different constructs were permeabilized and incubated at 37 °C for six minutes in a solution containing 2 mM ATP, 10 µM free Ca^2+^, and Alexa Fluor 488-tagged G-actin monomers (see Materials and Methods). After fixation, the fluorescence intensity of each cell was analyzed by confocal microscopy. Figure 5A shows epifluorescence images of three representative C25 myoblasts expressing de novo polymerized F-actin filaments (green), and alternatively WT, p.A618T or p.S619L dynamin-2 (red), and the DAPI nuclear signal (blue). At first glance, the decrease in new F-actin filaments in cells expressing the dynamin mutants is notorious. As a result, the percentage of F-actin de novo formation in the whole cytosol significantly decreased in cells expressing the mutations, as compared with the WT condition (Figure 5B). This decrease was also observed when the fluorescence intensity was quantified in the cortical area (Figure 5C). Then, as previously observed with other dynamin-2 mutations linked to CNM [21], p.A618T and p.S619L variants also impair actin dynamics.

## 3. Discussion

Dynamin-2 is a GTPase canonically involved in endocytosis; however, it has also been implicated in vesicle trafficking and exocytosis in different types of cells, including chromaffin cells [8], natural killer cells [11] and pancreatic β cells [12], among others. In skeletal muscle cells, exocytosis is a ubiquitous and essential mechanism that controls the surface expression of receptors, channels, and transporters, as well as the release of myokines with autocrine and paracrine effects [18,19]. Then, the malfunction of exocytosis might affect not only skeletal muscle cells but also other tissues. Indeed, the surface expression of GLUT4 receptors impacts the whole-body metabolism [40], and myokine secretion influences the physiology of different organs [41]. Here, we found that both Ca^2+^ and dynamin-2 influence exocytosis in skeletal muscle cells.

Cytosolic Ca^2+^ levels seem to importantly regulate exocytosis in skeletal muscle cells. Indeed, physical exercise induces GLUT4 exocytosis in part via cytosolic Ca^2+^ increments through a mechanism that involves Ca^2+^/calmodulin-dependent protein kinase II [42,43]. Skeletal muscle contractile activity further promotes ATP release through pannexin channels, with the consequent autocrine activation of purinergic P2Y receptors followed by Ca^2+^ release from intracellular stores through IP_3_ receptors [44] and GLUT4 insertion into the plasmalemma [45]. Insulin also promotes Ca^2+^ release from intracellular stores via IP_3_ and ryanodine receptors, contributing to insulin-induced GLUT4 translocation to the plasmalemma of myotubes and myofibers [43,46]. Here, by using TIRFM in human skeletal myoblasts, we monitored single fluorescence events of vesicles containing IRAP-pHluorin and found that the number of such events significantly augmented with the application of 5 and 10 μM ionomycin, as well as with the application of 100 nM insulin. The fact that an acidic extracellular solution quenched almost all the pHluorin signals indicates that they correspond to exocytosis events. This conclusion is also supported by the fact that high HEPES concentration retards the decay of fluorescence transients. To provoke such an effect, HEPES must enter the vesicle lumen supposedly through the fusion pore. This idea is further supported by the increased expression of GLUT4 at the plasmalemma upon ionomycin stimulation, as we previously showed in the human myoblast cell line RCMH [21] (see also Appendix A).

In endocrine cells, the amplitude and duration of the cytosolic Ca^2+^ signals regulate different steps of vesicle trafficking and exocytosis, including vesicle distribution, mobilization, docking and fusion with the plasmalemma [47]. They further determine the mode of exocytosis [47]. Indeed, high cytosolic Ca^2+^ concentrations favor “full-fusion” [48], a mode of exocytosis wherein the vesicle completely collapses in the plasmalemma, different from the kiss-and-run mode, where the fusion pore closes, and the vesicle is recovered without collapsing in the plasmalemma [49]. In the present work, we further found that high ionomycin concentrations do not only increase the number of pHluorin signals (Figure 1) but remarkably also prolong the duration of every single signal and increase dwell-time lengths (Figure 2). Depending on the mode of exo/endocytosis, the dwell time can report the duration of the fusion pore [36] or the time of residency of vesicle proteins at the plasmalemma before its internalization [35]. The duration of the pHluorin signal is also influenced by the decay time, a parameter that can report the vesicle retrieval and its subsequent reacidification after kiss-and-run [36] or the lateral diffusion of the vesicle protein after full-fusion [37]. The analysis of the evolution of the sigma parameter in time obtained from the Gaussian fit of each fluorescence distribution showed that non-lateral diffusion events prevail over the events with lateral diffusion (Appendix A). As the retention of vesicle membrane proteins at the site of fusion has been related to the kiss-and-run mode of exocytosis [50], it is probable that this form of exocytosis prevails in C25 cells. This idea is further supported by the fact that 100 mM HEPES in the extracellular solution significantly slowed the decay times of the IRAP-pHluorin signals (Appendix A). These findings are different from those reported in rat adipocytes stimulated with insulin, wherein the IRAP-pHluorin fluorescence decay appears to obey IRAP dispersal via lateral diffusion [51]. Then, these dissimilar IRAP-pHluorin behaviors might reflect the prevalence of different modes of exocytosis in adipocyte and skeletal muscle myoblasts. Thus, whereas full-fusion events might prevail in adipocytes, kiss-and-run events seem to predominate in myoblasts.

Dynamin-2 has been involved in exocytosis in different types of cells [8,11,12,15]. Deletion of three dynamin isoforms in pancreatic β cells impairs insulin granule exocytosis [52]. In human gut cells, dynamin inhibition reduces exocytosis [15]. In neurons, genetic or pharmacological inactivation of dynamins strongly impairs exocytosis of dense-core vesicles [53]. Here we found that overexpression of WT dynamin-2 reduced the amplitude of the Ca^2+^ signals induced by 10 μM ionomycin (Appendix A). It has been proposed that ionomycin induces cytosolic Ca^2+^ signals via different mechanisms, including Ca^2+^ release from intracellular stores and store-operated Ca^2+^ entry [54]. Then, one or both of these mechanisms might be altered by a chronic (24 h) overexpression of WT dynamin, which could impact both endocytosis of plasmalemma proteins and intracellular signals activated by endocytosis [55]. However, despite the reduced Ca^2+^ signals, the number of pHluorin events was not reduced (Appendix A), suggesting that the overexpression of WT dynamin-2 upregulates exocytosis and/or vesicle trafficking to the plasmalemma. In this respect, it has been reported that dynamins are required to organize and recover exocytosis sites by a mechanism that appears to favor the assembly of the t-SNARE syntaxin-1 at exocytosis sites [53]. Furthermore, vesicle trafficking and exocytosis are importantly dependent on the actin cytoskeleton, and dynamin-2’s action on exocytosis depends on the actin dynamics [8,21]. Skeletal muscle cells express the cytoskeletal β and γ actin isoforms. In general, the β isoform is present in stress fibers, cell-cell contacts, and contractile rings, while the γ isoform is typically expressed in the cell cortex, suggesting that the latter isoform is most probably involved in exocytosis and endocytosis [56]. However, although γ-actin ablation impairs insulin-stimulated glucose uptake in skeletal myofibers of growing animals, a functional redundancy between both cytoskeletal isoforms seems to explain the lack of effects of the specific ablation of β or γ actin in the glucose uptake in mature skeletal myofibers [57,58]. Dynamin-2 mutations associated with CNM, such as p.R369W and p.R465W, disturb the formation of actin dynamics and actin-dependent GLUT4 trafficking in myoblasts [21]. Then, it is possible that the disruption of actin filament formation observed by us in myoblasts expressing the p.A618T and p.S619L mutations might be responsible for reduced exocytosis in these cells.

The p.A618T and p.S619L mutations also modified parameters such as the duration, dwell time and decay time of IRAP-pHluorin signals, suggesting that they further influence postfusion stages of the exocytosis process, such as fusion pore lifetime and vesicle membrane internalization. In this respect, it has been reported in astrocytes that the pharmacological activation of dynamin favored fusion pore closure and vesicle internalization, whereas its pharmacological inhibition prevented vesicle internalization [17]. The constriction of the fusion pore by dynamin has also been reported in neuroendocrine cells [59], and in L6 myoblasts, dynamin-2 reportedly regulates GLUT4 internalization [60,61]. Then, it seems that the gain-of-function mutations p.A618T and p.S619L do not only reduce the number of exocytosis events in human myoblasts but also favor the closure of the fusion pore and the vesicle internalization, as observed with the dynamin activator Ryngo-1-23 [17]. This might impact the time of residency of proteins at the plasmalemma. As IRAP-pHluorin has been considered a reporter for GLUT4 trafficking [26], these findings suggest that these CNM-causing mutations might impair GLUT4 surface expression in skeletal muscle cells, as observed in myofibers of knock-in mice and biopsies from CNM patients, both carrying the dynamin-2 p.R465W mutation [21]. An imbalance between exocytosis and vesicle internalization in skeletal muscle cells might importantly disturb expression and residency time at the plasmalemma of functional proteins, impairing the skeletal muscle tissue physiology and contributing to the CNM disease. Thus, these findings contribute to a better understanding of the pathological mechanism involved in the severe clinical phenotype produced by these mutations.

## 4. Materials and Methods

### 4.1. Plasmids

IRAP-PHluorin was gently provided by Dr. J. Lippincott-Schwartz [25]. The WT dynamin-2 mCherry plasmids were constructed by GenScript Corporation (Nanjing, China) by cloning the rat dynamin-2 isoform aa (Genbank L25605; 97.59% identity with human dynamin-2 sequence) into an mCherry_pcDNA3.1(+) vector. The dynamin-2 mCherry mutations A618T and S619L were constructed by site-directed mutagenesis using the QuikChange II XL Site-Directed Mutagenesis Kit (Agilent Technologies, Santa Clara, CA, USA) (DYN2 isoform aa, sequence in).

### 4.2. Culture of Cell Lines and Transfection

The C25 cell line obtained from the immortalization of human cells from the Institut de Myologie (Paris, France) was established from a biopsy of the semitendinosus muscle of an unaffected individual [62]. They were cultivated at a density of 3 × 10^5^ cells/mL in 25 mm glass coverslips in a 4:1 Dulbecco’s modified Eagle minimal essential medium/199 medium (Sigma-Aldrich St. Louis, MO, USA) supplemented with 20% fetal bovine serum, 25 μg/mL fetuin (Sigma-Aldrich, St. Louis, MO, USA), 0.5 ng/mL basic fibroblast growth factor (Gibco BRL, Gaithersburg, MD, USA), 5 ng/mL epidermal growth factor (Gibco BRL, Gaithersburg, MD, USA), 0.2 μg/mL dexamethasone (Sigma-Aldrich, St. Louis, MO, USA), 5 μg/mL insulin (Eli Lilly Co., Indianapolis, IN, USA), 50 U/mL penicillin (OPKO, Santiago, Chile), and 100 µg/mL gentamicin (Gibco/Life Technology, Shanghai, China) and incubated at 37 °C in a 5% CO_2_ atmosphere until experimentation. For transfections, the culture medium was replaced with serum-free Opti-MEM medium (Gibco BRL, Gaithersburg, MD, USA) containing 1 µg DNA and 1.5 µL of Lipofectamine 2000 (Invitrogen, Carlsbad, CA, USA) and incubated for 20 min. After that treatment, cells were kept at 37 °C in a 5% CO_2_ atmosphere for 4 h, and later, 3 mL of the culture medium was added, and cells were kept at 37 °C in a 5% CO_2_ atmosphere for 24 h prior to experimentation.

### 4.3. Live-Cell Fluorescence Imaging

Measurement of single exocytotic events using TIRFM and pHluorin reporters was conducted as we previously reported [33]. Briefly, coverslips with C25 myoblasts expressing IRAP-pHluorin were placed in a chamber and perfused with a recording solution containing (in mM): 140 NaCl, 5.9 KCl, 1.2 MgCl_2_, 2 CaCl_2_, 10 D-glucose, and 10 Hepes-NaOH, pH 7.4. Live-cell imaging experiments were carried out on an inverted microscope (Eclipse Ti-E, Nikon, Tokyo, Japan) equipped with a Perfect Focus Unit (Nikon, Tokyo, Japan), a 60×/1.49NA Plan APO TIRF objective (Nikon, Tokyo, Japan) and two lasers of 488 nm (488-20LS, OBIS, Coherent, Santa Clara, CA, USA) and 561 nm (Coherent Compass 561-25 DPSS, Coherent Scientific, Adelaide, Australia). IRAP-pHluorin was excited by the 488 nm laser line, and time-lapse images were acquired at 3.3 Hz for 3 min with a digital camera (C11440, ORCA-FLASH 2.0; Hamamatsu Photonics, Hamamatsu City, Japan) controlled by the NIS-Element viewer 4.3 software (Nikon, Tokyo, Japan). To induce exocytosis, ionomycin (Invitrogen, Carlsbad, CA, USA) in concentrations of 1, 5 or 10 μM or insulin (Eli Lilly Co., Indiana, USA) at 100 nM were applied 30 s after initiating the recording. When indicated, stimulations were done in an acidic extracellular solution containing (in mM): 140 NaCl, 2,4 KCl, 2 CaCl_2_, 2 MgCl_2_, 10 glucose, 10 HEPES and 10 citric acid (pH = 5.5). For experiments with 100 mM HEPES, the NaCl concentration was adjusted to keep osmolarity constant. All the experiments were performed at room temperature (20 ± 2 °C).

### 4.4. Image Analysis

Image sequences were analyzed with the ImageJ software (1.49v, NIH, Bethesda, MD, USA) implemented with a macro to analyze single fluorescence spots. The image sequence was recorded in AVI format and analyzed with custom software made with the Python-based algorithm “Anaconda”. Briefly, single-vesicle exocytosis events were visualized as bright spots in the evanescent field of TIRF microscopy. Spots were automatically detected considering specific features: spot diameter ranging from 2 to 50 pixels and a change in fluorescence (ΔF) of at least 3 standard deviations above baseline (F_0_).

To quantify the parameters of the exocytosis events, such as duration, dwell time, and decay, we used the fluorescent profile over time of each exocytotic event. Then, these parameters were calculated using Python3 language with Numpy, SciPy, Uncertainties, Statistics, Sympy, and Math libraries. To determine the duration of the exocytosis, we set the point before the fluorescence increment as the starting point and the point of the fluorescence that returned to basal levels as the ending point. The difference between these values corresponds to the duration. The dwell time was measured as the difference between the two points. The first and second values correspond to the minor point between the peak and the starting and the peak and the ending point, respectively. The decay was obtained by fitting an exponential between the endpoint of the dwell time and the ending value of the event using the following formula: y = Ae(x − x_0_/τ) + y_0_, where A is calculated by the SciPy curve fit function, x is the ending point of the dwell time, x_0_ is the ending point of the event, y_0_ corresponds to the minimum “x” value of the input data, and τ is the decay obtained from the curve fitting.

### 4.5. Non-Lateral and Lateral Diffusion Event Analysis

Potential events were firstly identified by using a standard segmentation processing. As a result, we automatically determined two regions of interest (ROIa and ROIb) for each possible event across the duration of the video. In general, ROIb was twice greater than ROIa. Next, the fluorescence time profile was determined by extracting the mean in the ROI at each frame. Standard deviation profiles were calculated for every frame by flattening the gray-scale ROI and fitting a Gaussian function by the maximum likelihood estimation method. Secondly, transitory events were determined by visual inspection. In brief, by using the fluorescence and standard deviation profiles, we determined if the event occurred after the temporal baseline (30 s) and finished before the end of the recording (180 s). Once we determined the number of transitory events, all events were isolated from each other, and we graphed each one of them in two different thermal maps, one containing the fluorescence event at time 0 (when the event just appeared) and the other containing the temporal mean in a spatial window of the frame (showing the evolution in the space). Both thermal maps events included both ROIs “a” and “b”. Those events that overflowed the area delimited by the ROIb were considered lateral diffusion events.

### 4.6. Actin Filaments Formation

The formation of actin filaments in myoblast was performed as previously described [21,63]. Briefly, C25 cells previously transfected with WT, p.A618T or p.S619L dynamin-2-mCherry plasmids were incubated in KGEP buffer (139 mM K^+^ glutamate, 20 mM PIPES, 5 mM EGTA, 2 mM ATP-Mg^2+^, pH 6.9) containing 10 µM free Ca^2+^, 0.3 μM Alexa Fluor 488-G-actin conjugate (Thermo Fisher Scientific, Waltham, MA, USA) and 20 μM digitonin (Sigma-Aldrich St. Louis, MO, USA) at 37 °C for 6 min [21,63]. Ca^2+^ and EGTA concentrations to achieve 10 µM free Ca^2+^ at pH of 6.9 and 25 °C were calculated using the online software Ca-EGTA Calculator v1.2 (University of California, Davis, CA, USA; https://somapp.ucdmc.ucdavis.edu/pharmacology/bers/maxchelator/CaEGTA-NIST.htm (accessed on 19 July 2019)). Subsequently, cells were fixed with 4% p-formaldehyde (PFA), stained with 5 mg/mL 4,6-diamidino-2-phenylindole (DAPI) and visualized at the equatorial plane in a C1 Plus laser-scanning confocal microscope (Nikon, Japan), equipped with a 60× objective (NA 1.49; Nikon, Japan) using identical exposure settings for all cells. For the quantification of new actin filament formation in the cytosol and cortical areas, fluorescence intensity was acquired using a custom macro developed with the Image J software. This macro identified those regions in which the fluorescence signals had intensity values over 67 on the scale of grays (0 to 255) and therefore were considered significant fluorescence areas. To quantify the percentage of total and cortical fluorescence intensity in each cell, fluorescence areas were divided by the total cell area without considering the nucleus area or by a 1 µm diameter area circumferentially aligned with the plasma membranes, respectively. All images were analyzed using ImageJ software (1.49v, NIH, Bethesda, MD, USA).

### 4.7. Cytosolic Ca^2+^ Measurements

C25 cells seeded on 25-mm glass coverslips were incubated with 5 μM Fluo-4 AM for 25 min at 37 °C.

Subsequently, Ca^2+^ signals induced with 10 µM ionomycin were obtained in an inverted microscope (Eclipse Ti-E, Nikon, Tokyo, Japan) using a 40× objective. Ionomycin was administered 20 s after the start of recording and was maintained throughout the acquisition. Images were acquired using a cooled digital camera (ORCA-FLASH 2.0; Hamamatsu Photonics, Hamamatsu City, Japan) and NIS-Element Advanced Research 4.3 software (Nikon, Tokyo, Japan). For cells transfected with the mCherry-tagged plasmid containing WT dynamin-2, the mutation p.A618T or the mutation p.S619L, experiments were performed 24 h after transfections. Data are presented as ∆F/F_0_, where F_0_ and F are the background-subtracted fluorescence intensities recorded immediately before and after the addition of the agonist, respectively.

### 4.8. Statistics

For each condition, we analyzed 9–24 independent cells from at least three different cultures. The normality of data was checked using the Kolmogorov–Smirnov test. Statistical significance was determined by using analyses of variance (ANOVA), followed by the Bonferroni test. Statistical comparisons of cumulative curves were performed using the Kolmogorov–Smirnov test. Time constants (τ) were statistically analyzed using the t-test with Welch correction. Non-normally distributed data were compared with the Kruskal–Wallis test, followed by Dunn’s post hoc test. All statistical analyses were performed using GraphPad 8 software (GraphPad Software Inc, La Jolla, CA, USA). Results were expressed as means ± SEM for normally distributed data or in box plots for non-normal data, showing median (horizontal line) and max (top) and min (bottom) values of the distribution.

### 4.9. Ethics Statement

This research was approved by the Biosafety and Bioethics committees of Universidad de Valparaíso (Chile), with approval identification numbers BS002/2016 and BEA080-216, respectively.

## Figures and Tables

**Figure 1 ijms-23-10363-f001:**
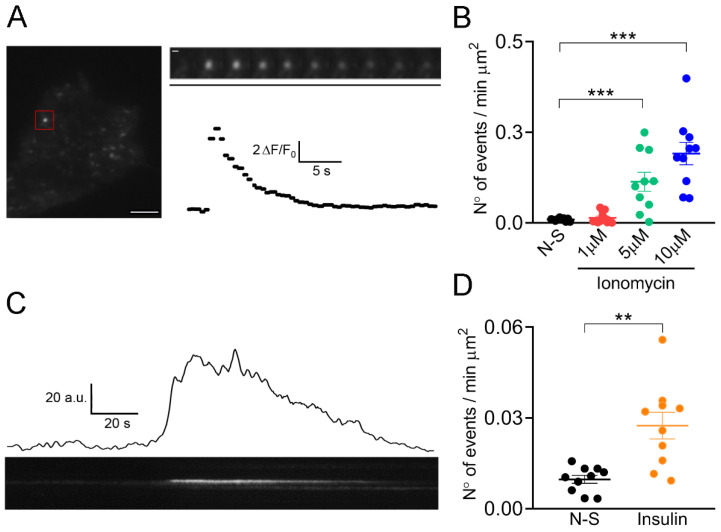
Spontaneous and stimulus-induced fusion events in immortalized human C25 myoblasts. C25 myoblasts were transfected with IRAP-pHluorin, and exocytosis was monitored 48 h later by TIRFM. (**A**): The left panel shows a TIRFM image showing spontaneous fusion events in a non-stimulated C25 cell. Scale bar = 10 μm. The red square indicates a selected fusion event. The sequence of video frames of this event is shown in the upper right panel (every other frame was skipped). Scale bar = 0.5 μm. Time scale bar (black line) = 5.58 s. The fluorescence intensity time progress profile of this event is shown below the frame sequence. (**B**): Frequency of events in C25 myoblasts non-stimulated (N-S; 10 cells) or stimulated with 1, 5 or 10 µM ionomycin (12, 10, and 10 cells, respectively). Dots represent values obtained from individual cells from at least three independent cultures. Horizontal lines indicate means with SEM, respectively. *** *p* < 0.001 compared to the non-stimulated condition (ANOVA test followed by Bonferroni multiple comparison test). (**C**): A typical example of a fluorescence trace (above) and kymograph (below) showing the evolution in time of a fusion event in C25 myoblasts, stimulated with 10 µM ionomycin. (**D**): Frequency of events in C25 myoblasts non-stimulated (N-S; 10 cells) or stimulated with 100 nM insulin (10 cells). Dots represent values obtained from individual cells from at least three independent cultures. Horizontal lines indicate means with SEM, respectively. ** *p* < 0.01 (Student’s *t*-test).

**Figure 2 ijms-23-10363-f002:**
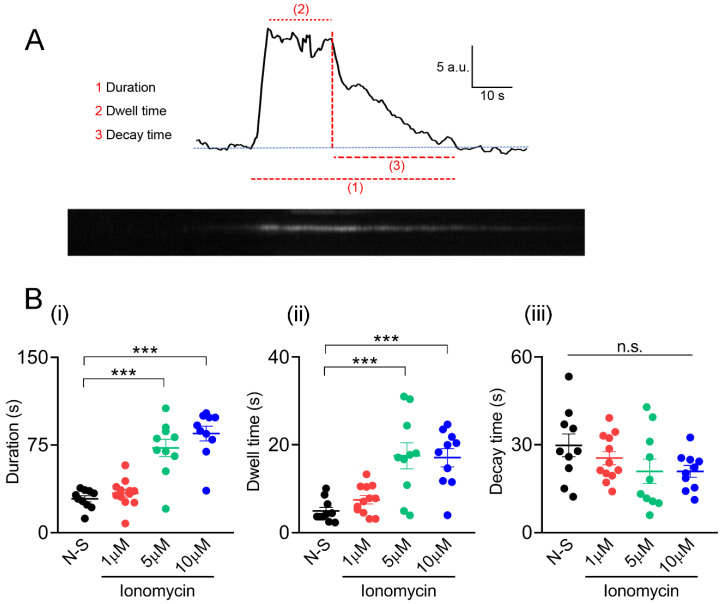
Kinetic parameters of spontaneous and ionomycin-induced pHluorin signals in C25 myoblasts. (**A**): An example of a trace of a transient fusion event induced in C25 myoblasts stimulated with 10 μM ionomycin (**top**) and its corresponding kymograph (**bottom**). The parameters analyzed were: (1) duration, (2) dwell time and (3) decay time. (**B**): A summary of these three parameters ((**i**), (**ii**) and (**iii**), respectively), obtained from pHluorin signals induced in non-stimulated (N-S; 10 cells) or stimulated C25 myoblasts with 1, 5, or 10 μM ionomycin (12, 10, and 10 cells, respectively), are represented. Dots represent values from individual cells from at least three independent cultures. Horizontal lines indicate means with SEM. N. *** *p* < 0.001, n.s. (non-significant difference) compared to spontaneous events (ANOVA test followed by Bonferroni multiple comparison test).

**Figure 3 ijms-23-10363-f003:**
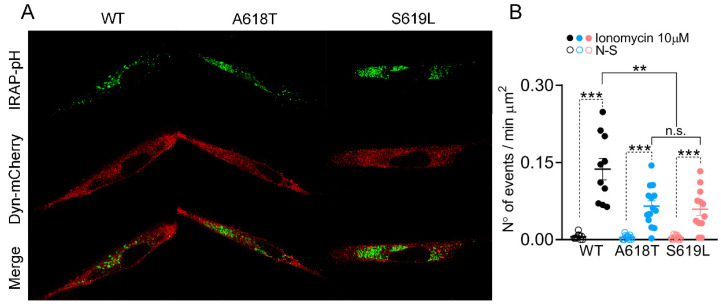
pHluorin signals in C25 myoblasts overexpressing WT dynamin-2 or the p.A618T or p.S619L mutations. C25 myoblasts were transfected with IRAP-pHluorin together with alternatively WT, p.A618T or p.S619L dynamin-2-mCherry plasmids. Exocytosis induced with 10 μM ionomycin was monitored 48 h later by TIRFM. (**A**): Confocal images of cells expressing IRAP-pHluorin (green) together with WT, p.A618T or p.S619L dynamin-2 (red). Scale bar = 20 μm. (**B**): Frequency of events in C25 myoblasts overexpressing WT (black; 10 cells), p.A618T (blue; 12 cells) or p.S619L (red; 14 cells) dynamin-2 in non-stimulated conditions (empty dots) or stimulated with 10 µM ionomycin (filled dots). Each dot represents the average frequency of events for individual cells from at least three independent cultures. Horizontal lines indicate means with SEM. *** *p* < 0.001 compared to the corresponding non-stimulated cells (Student’s *t*-test), ** *p* < 0.002 compared to WT dynamin-2 overexpressing cells stimulated with 10 μM ionomycin (ANOVA test followed by Bonferroni multiple comparison test). n.s., non-significant difference.

**Figure 4 ijms-23-10363-f004:**
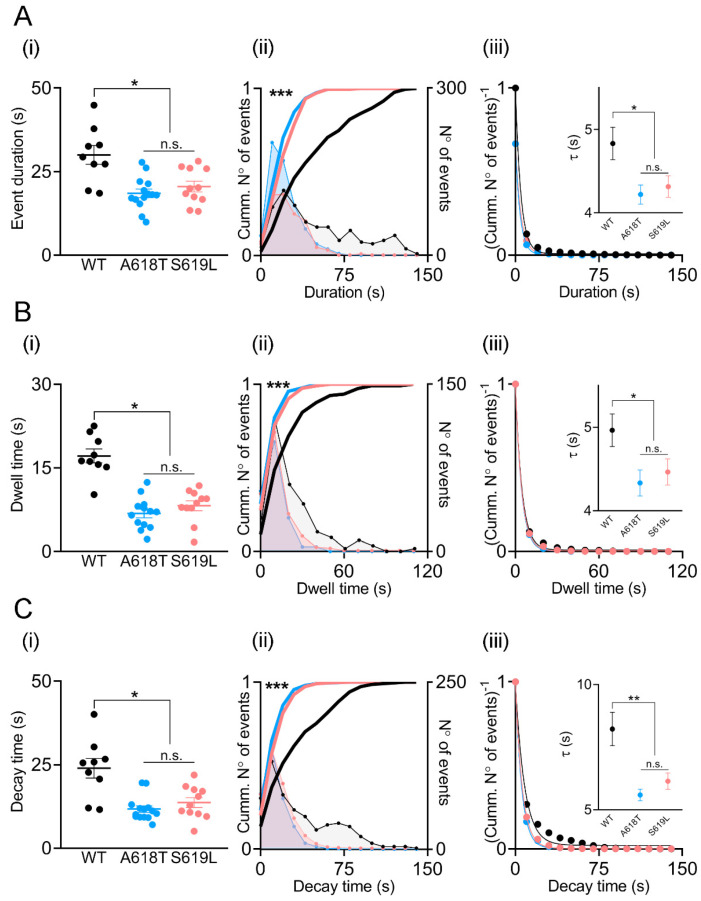
Kinetic parameters in C25 myoblasts overexpressing WT dynamin-2 or the p.A618T or p.S619L mutations. Analysis of event durations, dwell-time lengths and decay times in cells transfected alternatively with WT dynamin-2 (black), p.A618T (blue) or p.S619L (red). Panels (**i**) show average values of duration (**A**), dwell time (**B**), and decay time (**C**) from individual cells (filled dots) obtained from at least three independent cultures in each condition. Horizontal lines indicate means with SEM. The different conditions were compared by ANOVA test, followed by Bonferroni multiple comparison test * *p* < 0.05. Panels (**ii**) show the frequency distribution histograms (lines with dots; 10-s bin widths calculated using the Freedman–Diaconis rule) and its corresponding cumulative histograms (continuous lines) of events obtained during the recording period. The cumulative histograms of mutants vs. the WT condition were compared by the Kolmogorov–Smirnov test, *** *p* < 0.001. Panels (**iii**) show the inverse of the normalized cumulative distributions (i.e., survival curves), while insets show the time constants (τ) (mean ± SEM) obtained from their fitting to a single exponential decay function. Comparisons of both mutants vs. WT were performed by Student’s t-test with Welch’s correction, * *p* < 0.05, ** *p* < 0.01. n.s., non-significant difference.

**Figure 5 ijms-23-10363-f005:**
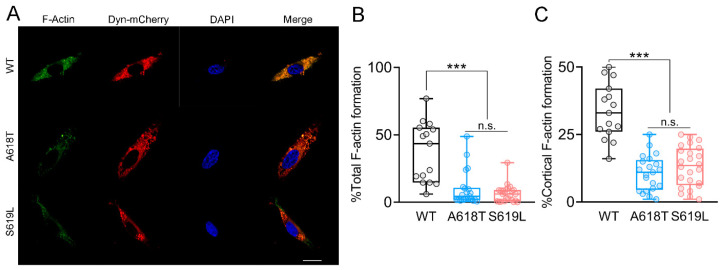
Actin filament formation in human myoblasts overexpressing WT dynamin-2 or expressing the p.A618T or the p.S619L mutations. C25 cells transfected with WT, p.A618T or p.S619L dynamin-2-mCherry plasmids were permeabilized with 20 μM digitonin in a KGEP buffer containing 2 mM ATP-Mg^2+^, 10 µM free Ca^2+^ and 300 nM Alexa-Fluor-488 actin for 6 min at 37 °C. After fixation, images were acquired at the equatorial plane of the cells by confocal microscopy using identical exposure settings for all cells. Panel (**A**) shows representative cells expressing de novo F-actin filaments (green), WT, p.A618T or p.S619L dynamin-2 (red) and the nuclear marker DAPI (blue). Scale bar = 20 µm. (**B**,**C**) Quantification of the fluorescence intensity of Alexa-Fluor-488 actin in the whole cytosol (**B**) and the cortical area (**C**). Data are expressed as the percentage of fluorescence area divided by the total cell area without nucleus area (**B**) or by the cell cortex area (**C**), according to described in methods. Horizontal lines and whiskers in box plots indicate medians and min (bottom) and max (top) values of the distribution. *** *p* < 0.001 compared to the WT condition (Kruskal–Wallis test followed by Dunn’s multiple comparison test). n.s., non-significant difference.

## Data Availability

The data presented in this study are available on request to the corresponding author.

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
