# Peer review of "Gain-of-Function Dynamin-2 Mutations Linked to Centronuclear Myopathy Impair Ca2+-Induced Exocytosis in Human Myoblasts"

_ijms, 2022, doi:10.3390/ijms231810363_

Round 1

Reviewer 1 Report

In this work, Bayonés et al., using Total Internal Reflection Fluorescence Microscopy (TIRFM) and the GLUT-4 vesicle fluorescence reporter (IRAP-pHluorin), describe the effect of dynamin-2 overexpression and two variants related to centronuclear myopathy in neonates; the mutations (p.A618T and p.S619L). This work is really interesting because it tries to show how these mutants affect the exo-endocytosis of GLUT-4 vesicles in a myoblast cell and give a functional explanation of some features of this disease.

However, I have some major criticisms and some issues need to be addressed by the Authors.

Major and general concerns:

- My main concern is related with the interpretation of IRAP-pHluorin signal. Multiple interpretations using this methodology are possible. It would be convenient to have some experiments to justify each of the stages described by the authors.

Usually, a fast rise in fluorescence (less than 1 s) indicates the moment of fusion of the vesicle with the plasma membrane. However, in Figure 1 C and more clearly in 1D the fluorescence rise is slow (almost 20 s). Perhaps, it only indicates the arrival of a vesicle (with neutral pH) to the plasma membrane.

When the authors define the dwell-time several questions arise.

Does the vesicle fusion pore remain open? Does the vesicle that remains in the plasma membrane become coated with clathrin?

Has been described that Dyn-1 and Dyn2 arrive to the vesicle just before departure of the vesicle and uncoating of clathrin. is it the same for mutated versions of dynamin-2?

In the decay time, the authors define it as the departure of the vesicle and reacidification. To check this, if this is the case, the authors should perform two-color TIRF to highlight the endocytic vesicle (e.g. with clathrin in red). Also, to highlight the movement of the vesicle in the z-plane, you should combine TIRF with epi-images time-lapse and calculate their ratio.

- I miss using insulin as a secretagogue in order to generate experimental conditions similar to those that occur in a physiological environment.

Concerns related with Figure 1

- IRAP-phluorin is a proved tool for monitoring GLUT-4 vesicle exocytosis by TIRFM. The co-distribution of IRAP-pHluorin with GLUT4-mCherry has been verified in adipocyte cells. In addition, the exocytosis of vesicles containing both proteins have been monitored after insulin stimulation. In a paper by Stenkula et al., 2010 describe “A rapid increase of the intensity of pHluorin was followed by a gradual decay associated with dispersal of IRAP molecules into plasma membrane. This was confirmed in individual cases by the analysis of intensity profile widening that corresponds to lateral diffusion”.

Stenkula et al., 2010 also describe using two-color TIRF microscopy two modes of GLUT4 exocytosis:fusion-with-release” or “fusion-with-retention”.

Stenkula et al., 2010 Cell Metab. Sep 8;12(3):250-9. doi: 10.1016/j.cmet.2010.08.005.

Insulin controls the spatial distribution of GLUT4 on the cell surface through regulation of its postfusion dispersal.

- There are differences using the same technique (IRAP-pHluorin and TIRFM) between the results published by Stenkula et al., 2010 and those described in this work that need to be contrast.

1.- The hallmark of exocytosis visualized by TIRF and probes based on EGFP or pHluorin result in both: an increase of fluorescence at the moment of fusion, because the change of pH, and a spread of signal (increases in s) due to the diffusion of the protein tagged with fluorescence protein. This have been well characterized using transmembrane proteins as transferrin receptor (TfR), sinaptobrevin (VAMP2) or even IRAP-pHluorin. On other hand, intravesicular proteins (depending on their molecular size) as tissue plasminogen activator (tpA), or chromogranin A (CgA) may be retained inside the vesicle after fusion and until the moment of endocytosis.

So, the observation that in “C25 cells expressing IRAP-pHluorin display fluorescence spots that do not diffuse laterally (see also kymographs in Figure 1 C and D)” must be accompanied by a plausible explanation.

2.- The authors categorize the events recorded as “transient” or "long-lasting" events according to their duration (shorter duration vs. longer duration). “The fusion events that had a shorter duration than the recording time were called “transient” events (Figure 1C), whereas other events that had longer durations were called “long-lasting” events (Figure 1D).” This criterion is ambiguous and depends very much on the duration of the recording and when the exocytosis event occurred, taking as a reference the onset of the stimulus.

3.- There is a significant difference in the mode of exocytosis recorded if we compare the decay time Fig. 1 B, iii of IRAP-pHluorin on average (25-30 s) with the results shown in Figure 5C of Stenkula et al., where the IRAP-pHluorin signal drops 1-2 s.

4.- The term fusion in this context is confusing and, in my opinion, should not be used. Fusion is well described and is only limited in exocytosis to the moment when the secretory vesicle fuses with the plasma membrane. This process, very well characterized in time by electrophysiological techniques, and its duration does not coincide with those described in Figure 1.

5.- The legend to fig. 1E indicates that “Bars show means ± SEM” and it is interpreted that mean ± SEM are calculated from cells from three different cultures, whereas in the description of the results of Fig. 1E it states the following: At non-stimulated and 1 µM ionomycin-stimulated conditions 77 ± 17 % of the 275 events and 67 ± 23 % of 596 of the events were transients, whereas in 5 and 10 µM ionomycin-stimulated conditions, 85 ± 25 % of 3759 events and 90 ± 8.3 % of 7394 were long-lasting events (Figure 1E), respectively. It appears that the mean and SEM are calculated from the total events. To me this is confusing and should be rewritten.

Figure 2

1.- Then, it seems that high cytosolic Ca2+ concentrations increase the duration of the fusion events by probably delaying IRAP-pHluorin retrieval. This statement does not reflect the results shown in Figure 2 and is speculative. Since the mechanism of action of ionomycin is well known, it is easy to assume that increasing concentrations of ionomycin can increase the cytosolic calcium concentration. I do not think it is appropriate, without the authors having measured cytosolic calcium under these experimental conditions, to establish a correlation between cytosolic calcium concentration and the residence time of IRAP-pHluorin in the plasma membrane.

The results shown are clear and interesting, but must be addressed in depth to establish the mechanism of action.

Figure 3

It has been shown that overexpression or mutations of dyn-2 can dysregulate the endosome recycling, causing an imbalanced expression of receptors, on the cell surface of the plasma membrane. in this regard, it must be said that the experiment described in Figure 3, Dynamin-2 is always overexpressed. 

1.- A comparison of IRAP-pHluorin exocytosis in unstimulated cells when dynamin-2 is or is not overexpressed is pertinent.

2.- On the other hand, it is also necessary to show a western-blot indicating the levels of overexpression of both WT and the two mutants.

3.-The size and quality of the image I have available to study the distribution of IRAP-pHluorin and Dynamin is not good. Please provide me with a larger and higher quality image.

Do dymanin-2 mutants have the same distribution pattern?

Supplemental Figure S1.

For this study it is important to characterize the cell model (human C25 myoblast) in relation to intracellular calcium mobilization. I have missed in the experimental design to include regular C25 myoblasts and a group transfected with an empty vector, also different concentration of ionomycin and others and more physiological secretagogues.

It is possible, if the transfection efficiency was not 100%, that cells without dynamin-2 overexpression were present in the same time-lapse that was recorded to analyze calcium dynamics.

Minor points:

- The following sentence of the A panel is confusing: A: Time-lapse obtained transients are shown as relative changes in fluorescence intensity (DF/F0). A suggestion in this regard would be:  Time-lapse images of the [Ca2+]i transients evoked by application of ionomycin are shown as relative changes in fluorescence intensity (DF/F0).

- In the Figure 1S B, if my interpretation is correct, each recording comes from a single cell and for quantitative analysis the highest fluorescence value referred as DF/F0 was taken. Therefore, each point represents the highest fluorescence value from a single cell and no as it was indicated Each dot represents averaged DF/F0 maximum values from individual cells from three different cultures”.

- In addition, the description of the graph is incorrect; the horizontal line inside the box may represent the mean. However, the whiskers do not indicate SEM.

- The description of this method should indicate that ionomycin (10 µM) was administered 20 s after the start of recording and was maintained throughout the acquisition. This observation is due to the fact that Figure 1S shows this time and yet the methods section specifies another time “To induce exocytosis ionomycin was applied 30 s after initiated the recording. All the experiments were performed at room temperature (20 ± 2°C)” and could lead to errors.

- Authors should be precise when describing the incubation time with Fluo-4 AM. There could be differences in the subcellular localization of Fluo-4 AM using incubation times of 30 or 45 minutes at 37°C.

Supplemental Figure S2.

- Your definition of non-lateral and lateral diffusion is incomplete probably because the method used by defining boxes of different sizes is not the most appropriate. The spread can be monitored on the line-profile by studying the sigma fit (s ) of the Gauss function over the time. In this sense, it has been reported in several publications Steyer and Almers 2001; Taraska and Almers 2004; Perrais et al., 2005 that the fluorescence from the granule can be fitted by Gaussian functions. The fit has several parameters (amplitude, width, center positions; x and y pixel positions.  The width (s) of the Gaussian function can be used as an estimate of the fluorophores spread during release from granules, even can be calculate the diffusion constant.

Minor points:

- The authors should describe that the image shows the fusion of two vesicles containing IRAP-pHluorin in a cell that also overexpresses dynamin-2 p.A618T. The image obtained in the red channel showing the distribution of dynamin-2 p.A618T should be displayed.

- It is described in the methods section that the image acquisition to perform the exocytosis studies was at 3.3 Hz. If so, the time scale shown in the middle of Figure 2A is not possible.

- The scale bar must be defined in the figure legend.

Author Response

We acknowledge the constructive and useful comments made by this reviewers, which greatly helped us in improving the manuscript.

Major and general concerns:

1.- My main concern is related with the interpretation of IRAP-pHluorin signal. Multiple interpretations using this methodology are possible. It would be convenient to have some experiments to justify each of the stages described by the authors. Usually, a fast rise in fluorescence (less than 1 s) indicates the moment of fusion of the vesicle with the plasma membrane. However, in Figure 1 C and more clearly in 1D the fluorescence rise is slow (almost 20 s). Perhaps, it only indicates the arrival of a vesicle (with neutral pH) to the plasma membrane.

Response: To determine whether the IRAP-pHluorin signals observed in our work correspond to exocytosis and not to the arrival of vesicles with neutral pH to the plasma membrane, cells were stimulated with 10 mM ionomycin in the presence of an acidic solution (pH=5.5). Reportedly, acidic solutions quench the pHluorin exposed to the extracellular space (Sankaranarayanan, 2000; doi: 10.1016/S0006-3495(00)76468-X). We found that only few fluorescence events were observed in the acidic condition, being less than 1% of the events observed with 10 mM ionomycin at pH 7.4 extracellular solution (p<0.005). A video is now showed as supplementary material (see in supplementary material video 2). To be sure that the acidic extracellular solution did not change the intraluminal pH of intracellular vesicles, cells expressing IRAP-pHluorin were visualized by confocal microscopy. The presence of fluorescent vesicles in a neutral extracellular solution (pH=7.4) indicated that some vesicles and organelles have non-acid intraluminal pH (see supplementary Figure S3). After changing the extracellular pH from 7.4 to 5.5, such vesicles continued being fluorescent (Figure S3). The analysis of 5 cells yielded Pearson's correlation coefficients of 0.85 ± 0.03.

We also performed experiments in the presence of 100 mM HEPES in the extracellular solution. They were performed to determine if the decay phase of the pHluorin signals depends on vesicle acidification. As shown in the supplementary figure S4, 100 mM HEPES slowed the decay time. In addition to support the idea that the decay time depends on the vesicle reacidification, these results further confirm that we are measuring exocytotic events, since HEPES, to exert its effects, had to diffuse through an opened fusion pore. These results are described in page 4 and discus in page 9.

2.-When the authors define the dwell-time several questions arise. Does the vesicle fusion pore remain open? Does the vesicle that remains in the plasma membrane become coated with clathrin?

Response: In the current manuscript we include additional interpretations of the dwell-time. In page 4, we now indicate that the dwell-time can report the residency of vesicle proteins at the plasmalemma before its endocytosis [Leitz and Kavalali, 2011, doi: 10.1523/JNEUROSCI.3358-11.2011] but has also may reflect the duration of the fusion pore before its closing (Vardjan et al. 2007, doi: 10.1523/JNEUROSCI.0351-07.2007). This is indicated in pages 4 and 9.

3.-Has been described that Dyn-1 and Dyn2 arrive to the vesicle just before departure of the vesicle and uncoating of clathrin. is it the same for mutated versions of dynamin-2?

Response: By using live correlative scanning ion conductance microscopy and fluorescence confocal microscopy, Ali et al. (doi: 10.1096/fj.201802635R) observed that both wild-type dynamin-2 and the p.R465W mutation are recruited to endocytic pit structures. They also demonstrated that this mutation disrupts the pit structure, and prevents its internalization (Ali et al., 2019). On the other hand, Srinivasan et al., 2016 (doi: 10.15252/embj.201593477) using TIRF microscopy observed that the dynamin-2 mutation p.S619L was not efficiently recruited to the plasma membrane in the presence of endogenous dynamin-2. However, after knockdown of the endogenous dynamin-2, the p.S619L mutant could be recruited to clathrin‐coated pits. The authors propose that the endogenous dynamin-2 competes with the p.S619L mutant for membrane‐binding sites.

4.-In the decay time, the authors define it as the departure of the vesicle and reacidification. To check this, if this is the case, the authors should perform two-color TIRF to highlight the endocytic vesicle (e.g. with clathrin in red). Also, to highlight the movement of the vesicle in the z-plane, you should combine TIRF with epi-images time-lapse and calculate their ratio.

Response: As aforementioned, to determine whether the decay time depends on vesicle reacidification we performed experiments using 100 mM HEPES in the extracellular solution. Reportedly, if fusion pore is wide enough for HEPES to enter the vesicular lumen, high HEPES concentrations slow down the reacidification of retrieved vesicles that underwent kiss-and-run (Vardjan et al. 2007, doi: 10.1523/JNEUROSCI.0351-07.2007). As shown in the new supplementary figure S4, 100 mM HEPES slow down the averaged decay-times, as well as tau values obtained from the single exponential fits. Then, these results support the idea that vesicle reacidification contributes to the fluorescence decay of the pHluorin signals. These results are complementary with the analysis of the evolution of the sigma parameter in time obtained from the Gaussian fit of each fluorescence distribution, as recommended by this reviewer. With this analysis we found that non-lateral diffusion events prevail over the events with lateral diffusion (see supplementary Figure S2). These results are described and discussed in pages 3, 4 and 9.

5.- I miss using insulin as a secretagogue in order to generate experimental conditions similar to those that occur in a physiological environment.

Response: We performed experiments with insulin in non-transfected cells. As show in the new Figure 1D, 100 nM insulin increased by 3.5-fold the event frequency in C25 cells.

6.-Concerns related with Figure 1.

6.1.-IRAP-phluorin is a proved tool for monitoring GLUT-4 vesicle exocytosis by TIRFM. The co-distribution of IRAP-pHluorin with GLUT4-mCherry has been verified in adipocyte cells. In addition, the exocytosis of vesicles containing both proteins have been monitored after insulin stimulation. In a paper by Stenkula et al., 2010 describe “A rapid increase of the intensity of pHluorin was followed by a gradual decay associated with dispersal of IRAP molecules into plasma membrane. This was confirmed in individual cases by the analysis of intensity profile widening that corresponds to lateral diffusion”. Stenkula et al., 2010 also describe using two-color TIRF microscopy two modes of GLUT4 exocytosis: “fusion-with-release” or “fusion-with-retention”. There are differences using the same technique (IRAP-pHluorin and TIRFM) between the results published by Stenkula et al., 2010 and those described in this work that need to be contrast.

Response: Yes, differently to the findings those observed by Stenkula et al. (2010) in rat adipose cells, we found in human skeletal myoblasts that only a small fraction of the IRAP-pHluorin signals spread out, whereas most of them exhibit non-lateral diffusion (see supplementary Figures S2). This issue is now discussed in pages 9-10, and we proposed that these differences might obey to different mode of exocytosis that occur in two distinct cell types (myoblasts and adipocytes) obtained from different species (human and rat).

6.2.- The hallmark of exocytosis visualized by TIRF and probes based on EGFP or pHluorin result in both: an increase of fluorescence at the moment of fusion, because the change of pH, and a spread of signal (increases in s) due to the diffusion of the protein tagged with fluorescence protein. This have been well characterized using transmembrane proteins as transferrin receptor (TfR), sinaptobrevin (VAMP2) or even IRAP-pHluorin. On other hand, intravesicular proteins (depending on their molecular size) as tissue plasminogen activator (tpA), or chromogranin A (CgA) may be retained inside the vesicle after fusion and until the moment of endocytosis. So, the observation that in “C25 cells expressing IRAP-pHluorin display fluorescence spots that do not diffuse laterally (see also kymographs in Figure 1 C and D)” must be accompanied by a plausible explanation.

Response: It has been proposed that the diffusion behavior of vesicle membrane proteins fused to pHluorin might reflect the mode of exocytosis (Tsuboi and Rutter, 2003, doi: 10.1016/s0960-9822(03)00176-3). In this regard, if the vesicle membrane protein is retained in the site of fusion (non-lateral diffusion) might indicate that the vesicle undergoes kiss-and-run, but it spreads out might reflect full-fusion. Then, it seems that kiss-and-run is the prevalent form of exocytosis in our cell model. This is idea is further support by the fact that 100 mM HEPES slows down the decay time of the pHluorin signals. This issue is now discussed in pages 9-10.

6.3.- The authors categorize the events recorded as “transient” or "long-lasting" events according to their duration (shorter duration vs. longer duration).The fusion events that had a shorter duration than the recording time were called “transient” events (Figure 1C), whereas other events that had longer durations were called “long-lasting” events (Figure 1D).” This criterion is ambiguous and depends very much on the duration of the recording and when the exocytosis event occurred, taking as a reference the onset of the stimulus.

Response: We agree with this reviewer. To now avoid this ambiguous criterion, we remove the definition of transient and long-lasting events, and the corresponding panels in Figure 1 and 3. Now, we only considered for our analysis the events that had a shorter duration than the recorded time.

6.4.- There is a significant difference in the mode of exocytosis recorded if we compare the decay time Fig. 1 B, iii of IRAP-pHluorin on average (25-30 s) with the results shown in Figure 5C of Stenkula et al., where the IRAP-pHluorin signal drops 1-2 s.

Response: We now compare our results with those of Stenkula et al. (2010). See pages 9-10. As we mention before, these differences could be a consequence of different mode of exocytosis that occur in two different types of cells.

6.5.- The term fusion in this context is confusing and, in my opinion, should not be used. Fusion is well described and is only limited in exocytosis to the moment when the secretory vesicle fuses with the plasma membrane. This process, very well characterized in time by electrophysiological techniques, and its duration does not coincide with those described in Figure 1.

Response: Considering this reviewer comment we now remove the term “fusion events” and replace it by “fluorescence events” or pHluorin signals.

6.6.- The legend to fig. 1E indicates that “Bars show means ± SEM” and it is interpreted that mean ± SEM are calculated from cells from three different cultures, whereas in the description of the results of Fig. 1E it states the following: At non-stimulated and 1 µM ionomycin-stimulated conditions 77 ± 17 % of the 275 events and 67 ± 23 % of 596 of the events were transients, whereas in 5 and 10 µM ionomycin-stimulated conditions, 85 ± 25 % of 3759 events and 90 ± 8.3 % of 7394 were long-lasting events (Figure 1E), respectively. It appears that the mean and SEM are calculated from the total events. To me this is confusing and should be rewritten.

Response: Panel E from figure 1 was now removed. The mean and SEM for that case were calculated from cells. We thank the reviewer for noticing the error.

7.-Figure 2. Then, it seems that high cytosolic Ca2+ concentrations increase the duration of the fusion events by probably delaying IRAP-pHluorin retrieval. This statement does not reflect the results shown in Figure 2 and is speculative. Since the mechanism of action of ionomycin is well known, it is easy to assume that increasing concentrations of ionomycin can increase the cytosolic calcium concentration. I do not think it is appropriate, without the authors having measured cytosolic calcium under these experimental conditions, to establish a correlation between cytosolic calcium concentration and the residence time of IRAP-pHluorin in the plasma membrane. The results shown are clear and interesting, but must be addressed in depth to establish the mechanism of action.

Response: We agree with this reviewer and therefore changed that sentence in the results section.

8.-Figure 3.

8.1.-It has been shown that overexpression or mutations of dyn-2 can dysregulate the endosome recycling, causing an imbalanced expression of receptors, on the cell surface of the plasma membrane. in this regard, it must be said that the experiment described in Figure 3, Dynamin-2 is always overexpressed. 

Response: According this reviewer suggestion we now indicate that dynamin-2 is overexpressed. This is indicated in red letters in page 6.

8.2.- A comparison of IRAP-pHluorin exocytosis in unstimulated cells when dynamin-2 is or is not overexpressed is pertinent.

Response: We now include a new supplementary figure (Figure S7) wherein we compare Ca2+ and pHluorin signals of non-transfected C25 myoblasts with those transfected with WT dynamin-2. We found that the overexpression of dynamin-2 reduced the amplitude of the Ca2+ signals, but not the number of fluorescence events. This issue is now discus in page 10.

8.3.- On the other hand, it is also necessary to show a western-blot indicating the levels of overexpression of both WT and the two mutants.

Response: The percentage of transfected cells in our conditions was very low: 2.2 ± 0.7, 3.5 ± 0.7 and 2.3 ± 0.7 % for WT dynamin-2 or the mutants A618T or S619L, respectively (see new supplementary Figure S5). In this regard, it is difficult to find differences in the expression by western blot.

8.4.-The size and quality of the image I have available to study the distribution of IRAP-pHluorin and Dynamin is not good. Please provide me with a larger and higher quality image.

Response: We now provide better images of cells expressing IRAP-pHluorin with the dynamin variants (see new Figure 3A).

8.5.-Do dymanin-2 mutants have the same distribution pattern?

Response:  As visualized by confocal microscopy using a 60x objective, all these dynamin-2 variants displayed a homogeneous cellular distribution, and only 8.7%, 12.5% and 9.1% of 23, 24 and 22 cells transfected with dynamin-2 WT, A618T or S619L, respectively, showed cytosolic aggregations. This information is included in page 5.

9.-Supplemental Figure S1.

9.1.- For this study it is important to characterize the cell model (human C25 myoblast) in relation to intracellular calcium mobilization. I have missed in the experimental design to include regular C25 myoblasts and a group transfected with an empty vector, also different concentration of ionomycin and others and more physiological secretagogues. It is possible, if the transfection efficiency was not 100%, that cells without dynamin-2 overexpression were present in the same time-lapse that was recorded to analyze calcium dynamics.

Response: We now include measurements of Ca2+ signals induced with 1, 5 and 10 mM ionomycin in non-transfected cells (supplementary Figure S1), as well as with 10 mM ionomycin in cells transfected with the empty vector (supplementary Figure S6).

9.2.-Minor points. The following sentence of the A panel is confusing: A: Time-lapse obtained transients are shown as relative changes in fluorescence intensity (DF/F0). A suggestion in this regard would be:  Time-lapse images of the [Ca2+]i transients evoked by application of ionomycin are shown as relative changes in fluorescence intensity (DF/F0).

Response: We appreciate this correction and have changed the sentence according to this reviewer suggestion.

9.3.- In the Figure 1S B, if my interpretation is correct, each recording comes from a single cell and for quantitative analysis the highest fluorescence value referred as DF/F0 was taken. Therefore, each point represents the highest fluorescence value from a single cell and not as it was indicated “Each dot represents averaged DF/F0 maximum values from individual cells from three different cultures”.

Response: We appreciate this correction and have rewritten the sentence according to this reviewer suggestion. We now indicate: “Each dot represents DF/F0 maximum values from individual cells from three different cultures”.

9.4.- In addition, the description of the graph is incorrect; the horizontal line inside the box may represent the mean. However, the whiskers do not indicate SEM.

Response: We appreciate this correction, and this was amended in the legends.

9.5.- The description of this method should indicate that ionomycin (10 µM) was administered 20 s after the start of recording and was maintained throughout the acquisition. This observation is due to the fact that Figure 1S shows this time and yet the methods section specifies another time “To induce exocytosis ionomycin was applied 30 s after initiated the recording. All the experiments were performed at room temperature (20 ± 2°C)” and could lead to errors.

Response: Thanks for noticing that error. We now indicate in Methods that for Ca2+ signal measurements, ionomycin was administered 20 s after the start of recording and was maintained throughout the acquisition.

9.6- Authors should be precise when describing the incubation time with Fluo-4 AM. There could be differences in the subcellular localization of Fluo-4 AM using incubation times of 30 or 45 minutes at 37°C.

Response:  The incubation time with Fluo-4 AM actually was 25 min. This is specified in the text now.

10.-Supplemental Figure S2.

10.1- Your definition of non-lateral and lateral diffusion is incomplete probably because the method used by defining boxes of different sizes is not the most appropriate. The spread can be monitored on the line-profile by studying the sigma fit (s ) of the Gauss function over the time. In this sense, it has been reported in several publications Steyer and Almers 2001; Taraska and Almers 2004; Perrais et al., 2005 that the fluorescence from the granule can be fitted by Gaussian functions. The fit has several parameters (amplitude, width, center positions; x and y pixel positions.  The width (s) of the Gaussian function can be used as an estimate of the fluorophores spread during release from granules, even can be calculate the diffusion constant.

Response: As suggested by this reviewer, to determine whether the events display non-lateral or lateral diffusion we analyzed the evolution of the sigma parameter in time obtained from the Gaussian fit of each fluorescence distribution in each frame for every single event. This is described in Methods (page 12) and analyses are shown in the supplementary Figures S2 and S8.

Minor points:

10.2.- The authors should describe that the image shows the fusion of two vesicles containing IRAP-pHluorin in a cell that also overexpresses dynamin-2 p.A618T. The image obtained in the red channel showing the distribution of dynamin-2 p.A618T should be displayed.

Response: We now show image obtained in the red channel in Figure S8.

10.3- It is described in the methods section that the image acquisition to perform the exocytosis studies was at 3.3 Hz. If so, the time scale shown in the middle of Figure 2A is not possible.

Response: Yes, the scale was wrong. Thanks for noticing. We now corrected the scale (see new Figure 1A).

10.4.- The scale bar must be defined in the figure legend.

Response: The scale bar is now defined in the figure legend.

Reviewer 2 Report

The manuscript written by Bayonés et al. analyses the effect of dynamin-2 mutations related to centronuclear myopathy on  Ca2+-induced exocytotic events and endocytosis in human myoblasts.

In this regard, the authors have directly measured single exocytotic events and their subsequent endocytosis using total internal reflection fluorescence microscopy (TIRFM) in immortalized human myoblasts expressing the pH-sensitive fluorescent protein (pHluorin) fused to the insulin-responsive aminopeptidase IRAP as reporter of the GLUT4 vesicle-trafficking.

The results presented are interesting, and the manuscript should be of interest to the readership of the International Journal of Molecular Sciences.

I really enjoyed the reading of the manuscript and have just minor comments:

- Could you explain why for transfection of human myoblast the rat dynamin-2 isoform was used. What is the identity level between rat and human dynamin-2 sequences? The information about NCBI Reference Sequence of used rat dynamin-2  should be added.

- Because the effect of the mutation in dynamin-2 on the dynamics of the cytoskeleton was analyzed, in the introduction it would be useful to show the information about actin isoforms involved in exo and endocytosis processes in the myoblasts.

- A graphical abstract of the results would make the work more attractive for readers.

Author Response

We thank the positive and constructive comments of this reviewer, which greatly helped us in improving the manuscript.

1.- Could you explain why for transfection of human myoblast the rat dynamin-2 isoform was used. What is the identity level between rat and human dynamin-2 sequences? The information about NCBI Reference Sequence of used rat dynamin-2 should be added.

Response: The identity level between rat and human dynamin-2 sequences is 97.59%. For this reason, we used these constructs that were available in our lab. The information of the dynamin-2 sequence used (DYN2 isoform aa, sequence in Genbank L25605) is now included in Methods.

2.- Because the effect of the mutation in dynamin-2 on the dynamics of the cytoskeleton was analyzed, in the introduction it would be useful to show the information about actin isoforms involved in exo and endocytosis processes in the myoblasts.

Response: We now include this information in the Discussion as following: Skeletal muscle cells express the cytoskeletal β and γ actin isoforms. In general, the β isoform is present in stress fibers, cell-cell contacts, and contractile rings, while the γ isoform is typically expressed in the cell cortex, suggesting that the latter isoform is most probably involved in exocytosis and endocytosis (Cárdenas et al., 2022, doi:10.32604/biocell.2022.019086). However, although γ‐actin ablation impairs insulin‐stimulated glucose uptake in skeletal myofibers of growing animals, a functional redundancy between both cytoskeletal isoforms seems to explain the lack of effects of the specific ablation of β or γ actin in the glucose uptake in mature skeletal myofiber (Knudsen et al., 2022, doi: 10.14814/phy2.15183; Madsen et al. 2018, doi: 10.1152/ajpendo.00392.2017)

3.- A graphical abstract of the results would make the work more attractive for readers.

Response:  A graphical abstract is now included.